# Protein Modification with Ribose Generates *N*^δ^-(5-hydro-5-methyl-4-imidazolone-2-yl)-ornithine

**DOI:** 10.3390/ijms23031224

**Published:** 2022-01-22

**Authors:** Ikuho Ban, Hikari Sugawa, Ryoji Nagai

**Affiliations:** Laboratory of Food and Regulation Biology, Department of Bioscience, Graduate School of Agriculture, Tokai University 9-1-1, Higashi-ku, Kumamoto 862-8652, Japan; b.ikh.bad.0218@gmail.com (I.B.); sh855655@tsc.u-tokai.ac.jp (H.S.)

**Keywords:** AGEs, glycation, ribose, MG-H1, *Trapa bispinosa* Roxb., MG, LC-QTOF

## Abstract

Advanced glycation end products (AGEs) are associated with diabetes and its complications. AGEs are formed by the non-enzymatic reactions of proteins and reducing sugars, such as glucose and ribose. Ribose is widely used in glycation research as it generates AGEs more rapidly than glucose. This study analyzed the AGE structures generated from ribose-modified protein by liquid chromatography–quadrupole time-of-flight mass spectrometry. Among these AGEs, *N*^δ^-(5-hydro-5-methyl-4-imidazolone-2-yl)-ornithine (MG-H1) was the most abundant in ribose-glycated bovine serum albumin (ribated-BSA) among others, such as *N*^ε^-(carboxymethyl) lysine, *N*^ε^-(carboxyethyl) lysine, and *N*^ω^-(carboxymethyl) arginine. Surprisingly, MG-H1 was produced by ribated-BSA in a time-dependent manner, whereas methylglyoxal levels (MG) were under the detectable level. In addition, *Trapa bispinosa* Roxb. hot water extract (TBE) possesses several anti-oxidative compounds, such as ellagic acid, and has been reported to inhibit the formation of MG-H1 in vivo. Thus, we evaluated the inhibitory effects of TBE on MG-H1 formation using ribose- or MG-modified proteins. TBE inhibited MG-H1 formation in gelatin incubated with ribose and ribated-BSA, but not in MG-modified gelatin. Furthermore, MG-H1 formation was inhibited by diethylenetriaminepentaacetic acid. These results demonstrated that ribose reacts with proteins to generate Amadori compounds and form MG-H1 via oxidation.

## 1. Introduction

The number of deaths caused by lifestyle-related diseases, such as cancer, heart stroke, and type 2 diabetes, is increasing worldwide [1]. The number of patients with type 2 diabetes is projected to reach 470 million by 2030 [2]. Diabetes is associated with serious complications, such as neuropathy, nephropathy, retinopathy, and arteriosclerosis; it is important to prevent their pathogenesis as treatment is difficult once they have progressed.

The amino residues of proteins react non-enzymatically with reducing sugars, such as glucose, via the Maillard reaction, to form advanced glycation end-products (AGEs) through irreversible reactions such as oxidation and condensation [3,4]. Chronic hyperglycemia in diabetes increases the formation of AGEs [5]. Furthermore, the accumulation of AGEs is enhanced by the pathogenesis of diabetic complications [6]. In fact, it has been reported that the levels of *N*^ε^-(carboxymethyl) lysine (CML) [7] and *N*^δ^-(5-hydro-5-methyl-4-imidazolone-2-yl)-ornithine (MG-H1) [8], one of AGEs, are increased in the plasma of patients with nephropathy [9]. Thus, inhibiting the formation of AGEs is hypothesized to prevent the progression of diabetic complications.

As ribose shows higher reactivity with proteins than glucose, it rapidly generates AGEs, which are widely studied in glycation research [10,11,12,13]. For example, Han et al. reported that intraperitoneal administration of ribose in mice induced spatial cognitive impairment via the activation of an astrocyte-mediated receptor for AGEs (RAGE) dependent inflammatory response by ribose-derived AGEs [14]. Lu et al. further reported that ribose and AGE levels in serum were significantly elevated in patients with type 2 diabetes mellitus (T2DM) (Control: 75.71 ± 1.83 μM, T2DM: 89.90 ± 2.40 μM) [15]. These reports suggest that ribose plays an important role in glycation *in vivo*. Nevertheless, a few studies have reported the formation pathway of ribose-derived AGE structures. In a previous study, we developed a ribose-based screening system for inhibitors of glyoxal (GO)-derived AGEs, such as CML [16]. This system is possible to evaluate the inhibitory effect on AGE formations more efficiently than glucose since ribose rapidly generates AGEs than glucose. However, whether methylglyoxal (MG)-derived AGEs such as MG-H1 are formed from ribose-modified proteins has not been confirmed yet. In this study, we analyzed the AGE structures generated from ribose-modified proteins and clarified the underlying pathways of their formation.

## 2. Results

To quantify the AGEs generated from ribose-derived BSA (ribose-BSA), the samples were analyzed by liquid chromatography–quadrupole time of flight mass spectrometry (LC-QTOF) (CML, MG-H1, *N*^ε^-(carboxyethyl) lysine (CEL), and *N*^ω^-(carboxymethyl) arginine (CMA)) or high-performance liquid chromatography (HPLC) with a fluorescence detector (pentosidine) (Figure 1a). We found that ribose-BSA generated not only the GO-derived AGEs CML and CMA, but also the MG-derived AGEs CEL and MG-H1 (Figure 1b). Pentosidine was also generated in ribose-BSA (Figure 1c). In addition, CMA, CML, CEL, and MG-H1 were normalized by protein contents to compare pentosidine, thus the data were expressed as pmol/μg protein (Table 1) [17].

Next, MG-H1 levels in Amadori-BSA derived from ribose (ribated-BSA) were measured to deduce the formation pathway. The generation of MG-H1 from ribated and glycated-BSA was confirmed (Figure 2a). MG-H1 was generated by ribated-BSA in a time-dependent manner (Figure 2b), and its yield in ribated-BSA was 9.9-fold higher than that in Amadori-BSA derived from glucose (glycated-BSA) (Figure 2b), while MG was undetectable (<25 nM) in ribated-BSA (Figure 2c,d). The amount of arginine in ribated-BSA (nmol) did not change with incubation time (Table 2).

Since *Trapa bispinosa* Roxb. hot water extract (TBE), which exerts anti-oxidative activity, inhibits the formation of MG-H1 in vivo [18], the inhibitory effect of TBE on MG-H1 formation from ribose-gelatin was evaluated by monoclonal anti-MG-H1 antibody [19] as ELISA is suitable for multiple samples such as in the screening of AGE inhibitors [16]. In addition, we incubated gelatin with ribose as the arginine content in gelatin was three-fold higher than that of BSA, and because it could easily generate MG-H1. As result, TBE inhibited MG-H1 formation in a dose-dependent manner (Figure 3a). In contrast, TBE did not inhibit MG-H1 formation in a mixture of MG and gelatin, while aminoguanidine (AMG), a dicarbonyl trapping reagent, inhibited the formation of MG-H1 during the incubation of MG with gelatin (Figure 3b). Finally, the inhibitory effect of TBE on MG-H1 formation by ribated-BSA was evaluated. Levels of MG-H1 in ribated-BSA were measured using LC-QTOF. As shown in Figure 3c, TBE (100 µg/mL) significantly inhibited MG-H1 formation by ribated-BSA (Figure 3c). Additionally, after incubation of ribose with BSA, MG-H1 and CML formation was significantly inhibited by diethylenetriaminepentaacetic acid (DTPA), a metal chelator (Figure 4). These results demonstrated that MG-H1 is formed by the oxidation of ribose-modified proteins.

## 3. Discussion

MG-H1 is known to be generated from MG, the degradation product of glyceraldehyde-3-phosphate in the glycolytic pathway [20], and its accumulation is enhanced by metabolic abnormalities [8]. It is also formed by the MG generated from the oxidation of glycated proteins [21]. Thus, glucose is thought to be the major precursor of MG-H1 formation. However, ribose is also deeply involved in AGE formations because of its higher reactivity with proteins compared to that of glucose. In fact, intraperitoneal administration of ribose in mice increased the glycated protein levels in the serum to a greater extent than that of glucose [10]. However, it is unclear whether ribose is also involved in MG-H1 formation. This study provides evidence that MG-H1 is generated by not only glucose but also ribose. Although CML is known to be generated from the reaction of ribose with proteins [10], in the present study the MG-H1 content in ribose-BSA was the highest among the AGEs determined by LC-QTOF. However, when AGE levels were normalized to that of proteins, CML level was higher than that of MG-H1. This change is dependent on the difference in the numbers of lysine and arginine in BSA, which were 59 and 23, respectively. Nevertheless, the level of CEL was the lowest among all the determined AGEs. We previously reported that arginine residues were modified by MG more preferentially than lysine residues [22]. Thus, we hypothesized that MG-H1 is formed by MG, which is generated from ribated-BSA as well as glycated-BSA. For this reason, the formation of MG and MG-H1 from ribated-BSA was also analyzed using LC-QTOF. Although MG-H1 levels in ribated-BSA were significantly increased, those of MG were below the detectable levels. In addition, the MG-H1 levels in ribated-BSA increased in accordance with the incubation period, whereas the amount of arginine (nmol) did not change. However, modification sites of Arg on BSA by ribose have not been identified. Further study is required to identify these modification sites. These results suggested that MG-H1 was formed directly from Amadori products on ribated-BSA, but not via MG. Since CML is formed by the oxidation of Amadori products [23], we postulated that MG-H1 could also be formed by the oxidation of ribated-BSA.

Accordingly, we investigated whether MG-H1 formation in ribated-BSA or ribose-modified proteins was inhibited by the antioxidative activity using TBE. Jinno et al. have reported that the daily administration of TBE in clinical trials inhibited MG-H1 levels in serum [18]. In addition, we previously reported that TBE contains multiple polyphenols, such as gallic acid, ellagic acid, and eugeniin [24]. In particular, ellagic acid has been suggested to inhibit CML formation by anti-oxidative activity, but not by carbonyl trapping [25]. Therefore, we evaluated the inhibitory effect of TBE on MG-H1 formation using ELISA and LC-QTOF. ELISA was performed using a monoclonal anti-MG-H1 antibody, which was correlated with LC-QTOF [19]. TBE inhibited MG-H1 formation in ribose-modified gelatin and ribated-BSA, but not in MG-modified gelatin. These results suggested that MG-H1 formation is inhibited by the antioxidant activity of ingredients such as ellagic acid in TBE. In addition to that of CML, MG-H1 formation in ribose-modified BSA was significantly inhibited by the addition of DTPA. Taken together, our conclusion that MG-H1 formation from Amadori products on ribated-BSA is based on two lines of evidence: (i) Formation of MG was less than detectable level (< 25 nM) after incubation of ribated-BSA (Figure 2c), and (ii) AMG did not inhibit the formation of MG-H1 from ribose-modified gelatin (Figure 3a).

Based on these results, we postulated the possible pathways for ribose-derived AGE formations (Figure 5).

Our LC-QTOF system detects 25 nM MG solution (0.125 pmol/5 µL injection), as shown in Figure 2d. Although the presence of ribose and 1,2-dicarbonyls were not measured after dialysis, we observed that formation of MG was below the detectable level (<25 nM) after incubation of ribated-BSA (Figure 2c). Otherwise, it is possible that the generated MG from ribated-BSA could not be detected due to the rapid reaction with Arg on BSA.

We also speculated that the MG-H1 is generated from ribose through CEA, and tried to measured CEA in ribose-BSA. However, some [^13^C_6_] CEA changes to MG-H1 even at room temperature (20–24 °C) which makes it difficult to quantify MG-H1 and CEA in modified proteins. Since Klöpfer et al. reported that MG-H1 is generated from CEA and MG-H3 [26], MG-H1 may be generated from ribose-BSA through CEA and MG-H3. 

Therefore, since we previously demonstrated that glucosone is a major intermediate during the incubation of glucose under oxidative condition [27], we speculated that ribosone may be generated from the incubation of ribose under oxidative condition. In fact, a clear *m/z* 271, speculated as DAN-ribosone adduct, spectrum was detected after incubation of ribated-BSA, suggesting that ribosone may play a role in MG-H1 formation during incubation of ribose with proteins. Additionally, since glucosone is generated from glucose under oxidative conditions [27], DTPA may decrease the formation of ribosone from ribose, resulting in lower MG-H1 levels. Previous studies have reported that the inhibition of AGE formations has beneficial effects. For example, we previously reported that mangosteen pericarp extract inhibits pentosidine formation in human serum and improves skin elasticity [28]. In addition, the co-administration of TBE and lutein to type 1 diabetic rats inhibited CML formation in serum and prevented the progression of cataractogenesis [24]. Moreover, oral administration of citric acid to diabetic rats reduces the accumulation of CEL in lens proteins [29]. Therefore, natural compounds such as citric acid and TBE play an important role in the inhibition of AGEs in diabetes, and the identification of natural products that have an inhibitory effect on AGE formations is important.

Although ribose level in blood is about one-fiftieth of glucose revel, the serum levels are increased in the pathogenesis of diabetes [15]. In addition, ribose plays an essential role in the pentose phosphate pathway in organs such as the kidney, liver, and adipose tissues, where it accumulates [30]. It has also been reported that the expression of RAGE and NF-κB in mesangial cells was increased by the addition of ribose [31]. Another study has demonstrated that MG-H1 is correlated with chronic kidney disease and nephropathy [32]. Thus, ribose-derived AGEs, including MG-H1, are involved in protein dysfunction associated with metabolic abnormalities in the kidney, liver, and adipose tissues. 

The present study demonstrated that MG-H1 generated by ribated-BSA was more than that by glycated-BSA. This study proves for the first time that ribose generates MG-H1 by reaction with proteins. This could be useful in future research on identifying the inhibitors of MG-H1 formation.

## 4. Materials and Methods

### 4.1. Measurement of AGE Contents in Ribose-BSA

D (−)-ribose (30 mM) (Fuji Film Wako Pure Chemical, Osaka, Japan) was mixed with 2 mg/mL BSA in 200 mM sodium phosphate buffer (pH 7.2) (NaPB) with/without 1 mM DTPA (Fuji Film Wako Pure Chemical, Osaka, Japan), and filtered using a sterile filter (ADVANTEC^®^, Tokyo, Japan). The 1 mL of mixture was then incubated at 37 °C for 7 days. After incubation, all the unreacted ribose in the mixture was removed by dialysis with a cellulose membrane (EIDIA Co., Ltd. Tokyo, Japan) at 4 °C for 12 h. The protein concentration was determined using the bicinchoninic acid (BCA) assay (Thermo Scientific, Waltham, MA, USA). The amounts of AGEs in ribose-BSA was measured by LC-QTOF (Bruker Daltonics, Bremen, Germany), as described previously [33]. Briefly, 50 µL of ribose-BSA (25 μg protein) in sodium borate buffer (0.1 M boric acid 1 mM DTPA, pH 9.1) was reduced with 0.1 M NaBH_4_ (Fuji Film Wako Pure Chemical, Osaka, Japan) including 5 mM NaOH at 25 °C for 4 h (1:1:0.1, *v*/*v*/*v*). To correct the variation of measured AGEs by pretreatment, standard 10 pmol [^2^H_2_] CML, [^2^H_3_] MG-H1, [^2^H_4_] CEL, (PolyPeptide Laboratories, Strasbourg, France) [^13^C_6_] CMA, 5 nmol [^13^C_6_] lysine, and [^13^C_6_] arginine (Cambridge Isotope Laboratories, Tewksbury, USA) were added to the samples, and were hydrolyzed in 6 M hydrochloric acid (HCl) at 100 °C for 18 h. Samples were dried in vacuo, resuspended in 1 mL distilled water, and passed over a Strata-X-C solid phase extraction column (Phenomenex, CA, USA) and eluted with 2 mL of 7% ammonia. The eluate was dried and resuspended in 200 µL of 20% acetonitrile containing 0.1% formic acid. After filtration using a 0.45 µm polytetrafluoroethylene membrane filter (Millipore, MA, USA), 5 µL of the sample was injected into the LC-QTOF system. Lysine-derived AGEs were normalized to the lysine content of the protein, and those derived by arginine were normalized to the arginine content; thus, the data were expressed as mmol/mol Lys (CML and CEL) or Arg (CMA and MG-H1). Pentosidine in the sample was measured by HPLC, as previously described [34].

### 4.2. Quantification of MG-H1 in Ribated-BSA or Glycated-BSA

A mixture of 1.6 M ribose or D (+)-glucose (Fuji Film Wako Pure Chemical, Osaka, Japan) and 50 mg/mL BSA in phosphate-buffered saline (PBS) containing 1 mM DTPA and 100 µM AMG (Fuji Film Wako Pure Chemical, Osaka, Japan) (1:1, *v*/*v*) was incubated at 37 °C for 5 days. To serve as control 50 mg/mL, BSA was incubated for the same duration. After incubation, the unreacted sugars in the mixture were removed by dialysis with a cellulose membrane in 200 mM NaPB at 4 °C for 12 h. The protein concentration was determined using BCA assay. Subsequently, 2 mg/mL BSA or Amadori-BSA was incubated in 200 mM NaPB at 37 °C over a course of 5 days (0, 3, and 5 days). After incubation, MG-H1 contents in samples were measured by LC-QTOF, as described in Section 4.1 “Measurement of AGEs contents in ribose-BSA”.

### 4.3. Detection of MG in Ribated-BSA

We measured MG formation in ribated-BSA and glycated-BSA using DAN (Tokyo Chemical Industry, Japan), as described previously [35,36]. Briefly, 2 mg/mL of ribated-BSA was incubated in the presence of 10 mM DAN in 200 mM NaPB at 37 °C for up to 5 days. Incubated samples were filtered using a 3000 molecular weight cut-off filter (12,000 rpm, 30 min). The filtered solutions (0.1 mL) were added to 0.4 mL of 20 mM citric acid buffer and passed over a Strata-X-C solid phase extraction column (Phenomenex, CA, USA). The amount of MG-DAN in the eluted samples with 2 mL of 7% ammonia and 75% acetonitrile was measured by LC-QTOF.

### 4.4. LC-QTOF Condition

LC was conducted on a ZIC^®^-HILIC column (2.1 × 150 mm, 5 µm; Merck Millipore, Billerica, MA, USA) that was maintained at 40 °C. The mobile phase was 0.1% formic acid (FA), with a two-step gradient of acetonitrile (ACN) (0–2 min, 90% ACN; 2–16 min, 90–10% ACN; 16–19 min, 10% ACN). The flow rate was set to 0.2 mL/min and the injection volume was 5 µL. The retention times for the four AGEs and amino acids were 12–15 min. The ionization source temperature was 200 °C, and the capillary voltage was 4.5 kV. Collision-induced dissociation was performed using nitrogen, with the collision energy set to 20 eV and pressure at 1.6 bar. Data were acquired with a stored mass range of *m*/*z* 50–1000. The composition formula of the detected ions was manually analyzed using the Smart Formula.

### 4.5. Inhibitory Effect of TBE on MG-H1 Formation by ELISA

Ribose (30 mM) was mixed with 2 mg/mL gelatin in 200 mM NaPB in the presence of AMG (0.1, 1, 10 and 100 µM) or TBE (0.01, 0.1, 1, 10, 100 µg/mL), and incubated at 37 °C for 7 days. MG-gelatin was prepared by incubating 2 mg/mL gelatin with 100 µM MG in PBS at 37 °C for 3 days in the presence of AMG or TBE. MG-H1 formation was measured by ELISA as previously described [37]. In brief, for noncompetitive ELISA, each well of a 96-well immune plate (Thermo Fisher Scientific, Waltham, MA, USA) was coated with 0.1 mL of the 1 µg/mL sample in PBS and blocked with 0.5% gelatin hydrolysate in PBS. The wells were incubated for 1 h with 0.1 mL of 1 µg/mL MG-H1 antibody [19]. Antibodies bound to the wells were detected using horseradish peroxidase-conjugated anti-mouse IgG antibody (Thermo Fisher Scientific, Waltham, MA, USA). Then, stained with 100 μL of 500 µg/mL *O*-phenylenediamine dihydrochloride (Fuji Film Wako Pure Chemical, Japan) in citrate-phosphate buffer (pH 5.0) containing 5.9 mM hydrogen peroxide for 3 min. The reaction was terminated with 100 μL of 1.0 M sulfuric acid, and the absorbance was measured at 492 nm using a Sunrise RAINBOW THERMO RC system (TECAN, Männedorf, Switzerland).

### 4.6. Inhibitory Effect of TBE on MG-H1 Formation in Ribated-BSA by LC-QTOF

Ribated-BSA was treated with 100 µg/mL TBE and incubated at 37 °C for 0 and 5 days. After incubation, the amount of MG-H1 in the samples were measured by LC-QTOF, as described the Section 4.1 “Measurement of AGEs contents in ribose-BSA”.

### 4.7. Statistical Analysis

All data are expressed as the mean ± standard deviation (SD). Figure 1a, Figure 2, Figure 3 and Figure 4 were examined for statistical significance using one-way analysis of variance with Bonferroni’s post hoc test. Statistical analyses were conducted using the EZR software package [38]. 

## Figures and Tables

**Figure 1 ijms-23-01224-f001:**
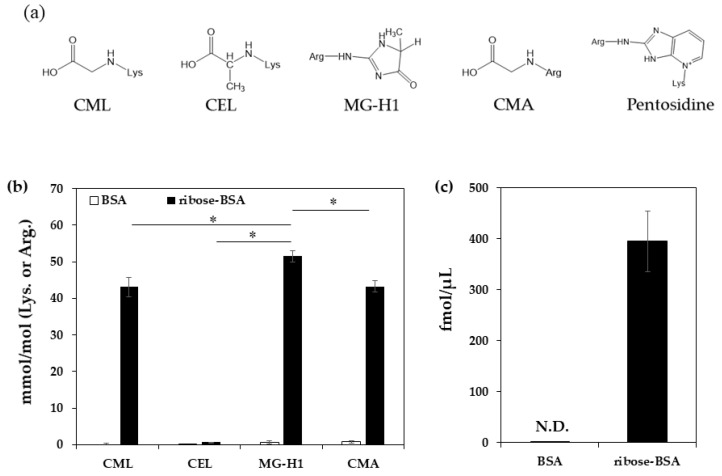
Formation of AGEs from ribose-BSA. The structures of AGEs detected from ribose-BSA (**a**). AGEs formed by ribose-BSA after seven days of incubation were quantified by LC-QTOF (*n* = 3); AGEs such as CML, CEL, MG-H1, and CMA (**b**). Pentosidine in ribose-BSA after seven days of incubation was quantified by HPLC (*n* = 3) (**c**). Data are presented as mean ± S.D., * *p* < 0.01, MG-H1 vs. CML, CEL, and CMA (Bonferroni test) (**b**).

**Figure 2 ijms-23-01224-f002:**
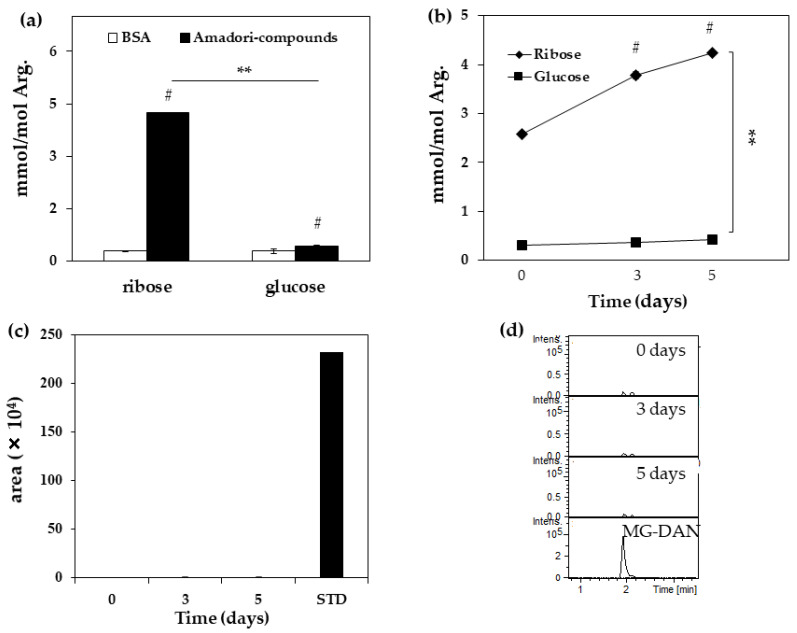
Formation of MG-H1 and MG from Amadori compounds. (**a**) Quantification of MG-H1 in ribated or glycated-BSA for five days. Open bars: Non-modified BSA, black bars: Amadori compounds (ribated or glycated-BSA) (*n* = 3); (**b**) Quantification of MG-H1 from ribated or glycated-BSA for zero, three, and five days (*n* = 3); (**c**) Measurement of MG from ribated-BSA for zero, three, and five days (*n* = 3) and 25 nM STD (0.125 pmol/5 μL injection).; (**d**) Chromatogram of MG-derived 2, 3-diaminonaphthalene (DAN) from ribated-BSA for zero, three, and five days by LC-QTOF. (*n* = 3). The data are presented as mean ± S.D. (**a**) ** *p* < 0.01, ribose vs. glucose (Bonferroni test). # *p* < 0.01, Amadori compound vs. BSA (Bonferroni test). (**b**) ** *p* < 0.01, ribose vs. glucose (Bonferroni test). # *p* < 0.01, three or five days vs. zero day (Bonferroni test).

**Figure 3 ijms-23-01224-f003:**
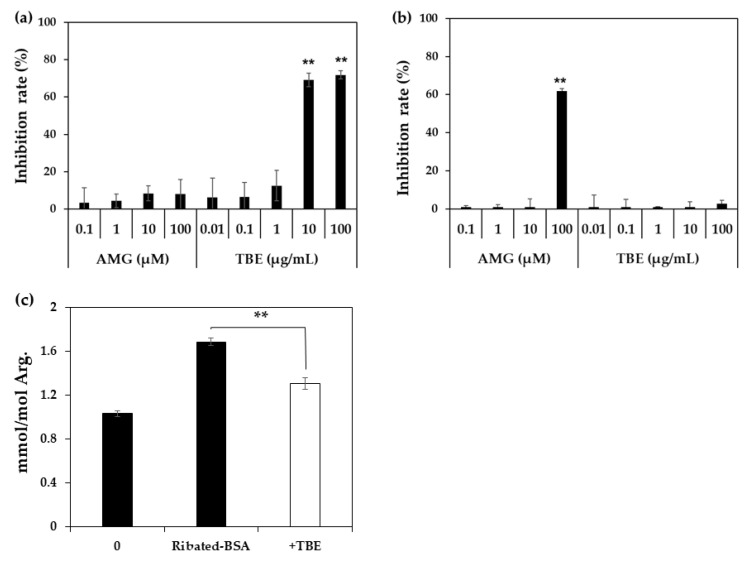
Inhibitory effect of TBE and AMG on MG-H1 formation. The inhibitory effect of AMG and TBE on MG-H1 formation from ribose-gelatin (*n* = 3) (**a**) or MG-gelatin (*n* = 3) (**b**) was evaluated by ELISA using an anti-MG-H1 antibody. Inhibitory effect of TBE on MG-H1 formation from ribated-BSA was evaluated using LC-QTOF (*n* = 3) (**c**). The data are presented as the mean ± S.D. (**a**,**b**) ** *p* < 0.01, vs. control (Bonferroni test). (**c**) ** *p* < 0.01, with TBE vs. without TBE control (Bonferroni test).

**Figure 4 ijms-23-01224-f004:**
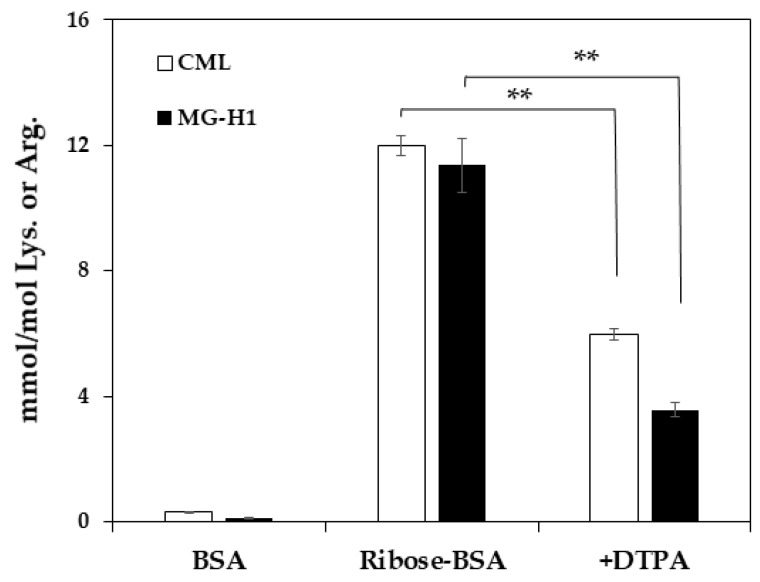
Formation of MG-H1 during incubation of ribose with BSA. To demonstrate the mechanism of MG-H1 and CML formation, ribated-BSA was incubated with/without DTPA, followed by LC-QTOF analysis (*n* = 3). The data are presented as the mean ± S.D., ** *p* < 0.01, ribose-BSA vs. without DTPA (Bonferroni test).

**Figure 5 ijms-23-01224-f005:**
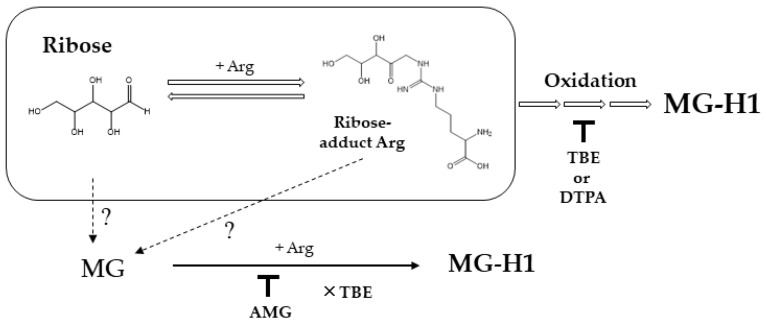
Scheme showing the pathway underlying AGE generation from ribose. Open arrow shows pathways on MG-H1 formation from reaction of ribose and Arg. Dot lines represent possible pathways. Down tack indicates inhibition by compounds, and cross mark in front of TBE shows no inhibitory effect.

**Table 1 ijms-23-01224-t001:** Levels of AGEs in ribose-BSA.

pmol/μg Protein	BSA	Ribose-BSA
CML	0.00 ± 0.31	21.20 ± 0.76
CEL	0.04 ± 0.00	0.33 ± 0.02
MG-H1	0.20 ± 0.13	13.06 ± 0.73
CMA	0.24 ± 0.11	10.95 ± 0.46
pentosidine	N.D.	0. 40 ± 0.06

Measurement of CML, CEL, MG-H1, CMA, and pentosidine generated from ribose-BSA (*n* = 3). Measurement by LC-QTOF and HPLC. The data are presented as the mean ± SD.

**Table 2 ijms-23-01224-t002:** The modification of Arg residues on ribated-BSA.

	Arg (nmol)	Change of Arg (%)
BSA	27.6	0
Ribated-BSA		
Day 0	20.0	27.8
Day 3	19.9	28.0
Day 5	19.6	29.2

Arg content was measured by LC-QTOF and normalized to protein content measured by BCA protein assay.

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
