# Peer review of "Protein Modification with Ribose Generates N?-(5-hydro-5-methyl-4-imidazolone-2-yl)-ornithine"

_ijms, 2022, doi:10.3390/ijms23031224_

Round 1
Reviewer 1 Report
Comments:
- How to confirm the modification of Arg residues on ribated-BSA?
- Authors could show the MALDI-TOF MS spectrum to confirm numbers of Arg residues on ribated-BSA.
- Do authors understand the modification sites (or modification sequence) of Arg residues on BSA?
- In Figure 4, authors could show the structures of detail structures of CML, CMA, MG-H1, Pentasidine and CEL. These structures could help readers to understand the results.
- How to avoid AGEs in diabetes? Authors could dissusion the issue clearly.
Author Response
We thank the reviewer for valuable comments, quoted in italics below, concerning our paper.
Concerns:
- How to confirm the modification of Arg residues on ribated-BSA?
Response:
The number of Arg residues on ribated-BSA was calculated based on Arg content, measured by LC-QTOF, of native BSA and ribated-BSA. Since native BSA contains 23 Arg residues, the number of Arg residues on ribated-BSA was calculated using the following formula.
Number of Arg residues on ribated-BSA = (mol of Arg on ribated-BSA) x (23/ mol of Arg on native BSA)
However, since Arg residues on unmodified BSA are also slightly modified in vivo, Arg content was expressed more precisely using “mol of Arg/ microgram protein” in the revised manuscript.
Table 2 legend: Arg content was measured by LC-QTOF and normalized to protein content measured by BCA protein assay. (Page 4, line 96)
Authors could show the MALDI-TOF MS spectrum to confirm numbers of Arg
Response:
Based on this comment, chromatograms of Arg as well as Lys are now shown in supplementary figure 1of the revised manuscript.
- Do authors understand the modification sites (or modification sequence) of Arg residues on BSA?
Response:
The modification sites of Arg on BSA are not known. Since the present study demonstrated that MG-H1 was generated from methylglyoxal as well as ribose, we focused on the quantification of MG-H1 by LC-QTOF, not MALDI-TOF MS, on modified BSA. Based on this comment, the following description has been added to the revised manuscript.
Discussion section:
However, modification sites of Arg on BSA by ribose have not been identified. Further study is required to identify these modification sites. (Page 5, line 150-152)
- In Figure 4, authors could show the structures of detail structures of CML, CMA, MG-H1, Pentasidine and CEL. These structures could help readers to understand the results.
Response:
We agree to this comment. AGE structures are shown in Fig. 1a of the revised manuscript.
- How to avoid AGEs in diabetes? Authors could dissusion the issue clearly.
Response:
We previously demonstrated that oral administration of citric acid to diabetic rat reduces the accumulation of CEL in lens proteins (Nagai et al., Biochem. Biophys. Res. Commun. 393, 118-122, 2010). Furthermore, the levels of CML and CEL were significantly inhibited when trapa bispinosa Roxb. and lutein were administered to diabetic rat (Kinoshita et al., J Clin Biochem Nutr. 66(1): 8-14, 2019), demonstrating that natural compounds such as citric acid and trapa bispinosa Roxb. play an important role in the inhibition of AGEs in diabetes. Based on this comment, the following description was added to the revised manuscript.
Discussion section:
Moreover, oral administration of citric acid to diabetic rats reduces the accumulation of CEL in lens proteins [29]. Therefore, natural compounds such as citric acid and TBE play an important role in the inhibition of AGEs in diabetes, and the identification of natural products that have an inhibitory effect on AGE formations is important. (Page 7, line 202-205)
Reviewer 2 Report
The reaction between reactive amino acid side chains (such as lysine or arginine) and reducing sugars leads to the formation of Amadori products. Further reaction leads to the formation of advanced glycation end-products (AGEs), which are covalently bound protein modifications. AGEs are of particular importance, because they are associated with the pathogenesis of especially oxidative-related diseases such as diabetes, cataract, or cardiovascular diseases. Thus, AGEs play an important role as (bio) marker compounds. To date, investigations of AGE formation are often based on glucose or other glycating agents such as 1,2-dicarbonyls.
Ban and colleagues investigate the formation of AGEs from ribose and focused on Nδ-(5-hydro-5-methyl-4-imidazolone-2-yl)-ornithine (MG-H1), Nε-(carboxymethyl) lysine (CML), Nε-(carboxyethyl) lysine (CEL), and Nω-(carboxymethyl) arginine (CMA). Interestingly, MG-H1 was the most abundant AGE among the studied. Ban et al. could show that MG-H1 is not only generated by methylglyoxal (MG) or glucose as described in previous work (e.g. Henle et al. Z Lebensm Unters Forsch (1994) 199:55-58), but also from ribose. For this, bovine serum albumine (BSA) was incubated with ribose under different conditions. Selected AGEs were analyzed by LC-QTOF analysis after acidic hydrolysis. This is an important finding, which is of interest to researchers of different fields (e.g. clinics, foods, biology, etc.). Overall, I think this study is publishable but there are some issues that must be addressed through revision. I have some questions / suggestions for the authors:
1. First, I would like to address some issues related to the discussion of the proposed formation pathway.
Finally, Ban et al. propose an oxidative formation pathway directly from the Amadori product. I cannot, however, follow this argumentation completely. Glomb and coworkers suggested formation of MG-H1 via MG-H3 and carboxyethylarginine (CEA) (J. Agric. Food Chem., Vol. 59, No. 1, 2011), which was confirmed by Frolov et al. (doi.org/10.1021/jf4050183). This formation pathway includes oxidation from MG to CEA (see Figures 6B from Frolov et al.). I am wondering why CEA was not investigated or mentioned, since it is an important intermediate product that leads to MG-H1. I suggest to also look at CEA levels, if possible. Investigating the CEA levels could help to draw a more detailed picture of the reaction mechanism of MG-H1 from ribose.
Can you exclude that MG-H1 is formed from other AGEs (e.g. earlier formed ones like MG-H3 or CEA) during the incubation of the ribated samples?
Formation of MG:
The amount of MG was investigated in ribated-BSA samples at different time points. Small molecules such as ribose and potential 1,2-dicarbonyls were removed by dialysis before incubation. Did you check whether ribose was (almost) completely removed by the dialysis process? Can you exclude that minor amounts of MG are formed during incubation of ribated-BSA e.g. from remaining ribose or Amadori products?
I think, that it is important to report the limit of detection and limit of quantification of the analytical method for MG detection. Without this information it is difficult to conclude on these experiments.
Experiments with DTPA:
There is a study by Gensberger-Reigl et al. that reported decreased MGO formation in glucose containing peritoneal dialysis fluids when DTPA was present (doi.org/10.1007/s10719-020-09964-6). Could it be possible that the presence of DTPA leads to a reduced formation of MGO and thus, to a lower amount of MG-H1? How can you differentiate between reduced MGO formation (which might lead to lower MG-H1 levels) and direct formation of MG-H1 from the Amadori product in the ribose-BSA samples with DTPA? Why was MG analyzed in ribated-BSA, but not in ribose-BSA samples?
In general, a reaction scheme that shows the chemical structures would be very helpful to follow the argumentation of the mechanistic interpretation of MG-H1 formation from ribose / the Amadori product.
2. Line 34: dehydration reactions are not generally irreversible, they are rather reversible and can be irreversible in particular situations. Please express this in a more differentiated way.
3. Lines 47/48: You mention that ribose plays an important role in glycation in vivo based on the work from Lu et al. and Han et al.. At this point, it would be helpful for the reader to get the information, which is given in lines 171/172. Alternatively, information about the actual ribose content in serum of T2DM patients would help.
4. Figure 1:
- Why did you use different units for the y-axis of 1a and 1b? I suggest normalizing the pentosidine content as well for better comparison.
- The error bar is missing in 1b BSA.
5. Line 76/77: "MG-H1 was generated by ribated-BSA in a time-dependent manner (Figure 2a)". There is no time curve displayed in Figure 2a.
6. Lines 79/80: "The amount of arginine in ribate-BSA (modification ratio) did not change with incubation time (Table 2)."
In the heading to Table 2 it is written that Arg residues on ribated-BSA are reported; in the table, however, ribose-BSA is mentioned.
What do you mean by "modification rate" and how is it calculated?
I do not understand the statement in lines 79/80. How did you determine the "amount of Arg" and how can the numbers (?) of Arg residues change during incubation experiments? After incubation, BSA will still have 23 Arg residues, some of them will be modified, some of them probably not. Do the numbers of Arg residues in Table 2 represent the number of modified or unmodified Arg residues? How did you analyze the number of Arg residues? Please clarify this. It might be better to display the amount of free Arg in the samples before and after incubation analyzed by LC-QTOF (see section 4.1).
According to Uniprot entry P02769 BSA usually has 26 Arg residues. Where does this difference (26 vs. 23) come from?
7. Figure 2:
In chapter 4.2 only the quantification of MG-H1 is mentioned, while the amount of Amadori-compounds is displayed in Figure 2a (black bars). In addition, only MG-H1 is mentioned in the respective figure caption. Thus, I assume that the black bar represents the amount of MG-H1? Please adjust the figure legend or, if my assumption is wrong, describe the analysis of the Amadori-compounds (e.g. in the Materials and Methods section) and the results more precisely. A more detailed figure caption to Figure 2a will also help (what was quantified - MG-H1 or Amadori-compounds / sum of selected AGEs)?
How can you explain the difference (factor 10³) in concentration levels between the results that are displayed in Figure 2a and 2b (nmol/mol Arg vs. mmol/mol Arg) - if Figure 2a displays the MG-H1 content (see question above)?
8. Lines 95ff: The inhibitory effect of TBE on the MG-H1 formation in BSA is explained and referred to the results displayed in Figure 3a. In the figure caption to Figure 3a, it is stated that the results were obtained on a ribose-gelatin system. This is inconclusive. Which system was used for the experiment displayed in Figure 3a?
Please give the concentration units for TBE and AMG in Figures 3a and 3b.
I think that it should be mentioned here, why gelatin was used instead of BSA, although the authors explain this in the discussion (lines 154/155). Otherwise, the reader cannot understand, why the protein was replaced at this point.
9. Lines 114/115: it seems that these lines belong to the figure cation of Figure 4.
10. Please introduce abbreviation at their first use (e.g. CMA, DAN, AMG, NaPB). Does NaPB stand for sodium phosphate buffer? If so, the term "buffer" doubled in lines 214, 215, 223, 240.
11. Please comment on the stability of AGEs during acidic hydrolysis.
12. Please review line 302ff. The consecutive numbering is interrupted.
13. Figure 5: "Pentsidine" should be "Pentosidine".
14. Line 87: please give the substance concentration (amount of substance / volume) of the MG standard.
Author Response
We thank the reviewer for the compliments and valuable comments, quoted in italics below, concerning our paper.
Concerns:
- Glomb and coworkers suggested formation of MG-H1 via MG-H3 and carboxyethylarginine (CEA) (J. Agric. Food Chem., Vol. 59, No. 1, 2011), which was confirmed by Frolov et al. (doi.org/10.1021/jf4050183). This formation pathway includes oxidation from MG to CEA (see Figures 6B from Frolov et al.). I am wondering why CEA was not investigated or mentioned, since it is an important intermediate product that leads to MG-H1. I suggest to also look at CEA levels, if possible. Investigating the CEA levels could help to draw a more detailed picture of the reaction mechanism of MG-H1 from ribose.
Can you exclude that MG-H1 is formed from other AGEs (e.g. earlier formed ones like MG-H3 or CEA) during the incubation of the ribated samples?
Response:
Although we agree with this comment, we learned that the precise quantification of CEA is difficult. We previously demonstrated that the monoclonal anti-CEA antibody recognizes MG-modified protein (9th International Symposium on the Maillard reaction, Germany, 2007). Although we also tried to measure CEA by LC-QTOF with the internal standard of CEA, which was synthesized from [13C6] arginine (Cambridge Isotope Laboratories, Inc), some [13C6] CEA changes to MG-H1 even at room temperature (20-24°C) which makes it difficult to quantify MG-H1 and CEA in modified proteins. Therefore, we measured CML, CMA and pentosidine in ribated-BSA as our preliminary experiment demonstrated that those standards do not change to MG-H1. Based on this comment, following description was added to the revised manuscript.
Discussion section:
We also speculated that the MG-H1 is generated from ribose through CEA, and tried to measured CEA in ribose-BSA. However, some [13C6] CEA changes to MG-H1 even at room temperature (20-24°C) which makes it difficult to quantify MG-H1 and CEA in modified proteins. Since Klöpfer et al reported that MG-H1 is generated from CEA and MG-H3 (J. Agric. Food Chem., Vol. 59, No. 1, 2011), MG-H1 may be generated from ribose-BSA through CEA and MG-H3. (Page 6, line 185-189)
- Formation of MG:
The amount of MG was investigated in ribated-BSA samples at different time points. Small molecules such as ribose and potential 1,2-dicarbonyls were removed by dialysis before incubation. Did you check whether ribose was (almost) completely removed by the dialysis process? Can you exclude that minor amounts of MG are formed during incubation of ribated-BSA e.g. from remaining ribose or Amadori products?
I think, that it is important to report the limit of detection and limit of quantification of the analytical method for MG detection. Without this information it is difficult to conclude on these experiments.
Response:
Since MG is generated from the oxidation of glycated protein [20] (page 5, lines 125-126 in original manuscript), we hypothesized that MG-H1 is formed by MG, which is generated from ribated-BSA as well as glycated-BSA (page 5, lines 139-140 in original manuscript). Therefore, as this reviewer noted, a special attention was paid to detect the formation of MG from ribated-BSA. Our LC-QTOF system detects 25nM MG solution (0.125 pmol MG/5 ml injection) as shown in Fig. 2d. Although the presence of ribose and 1,2-dicarbonyls were not measured after dialysis, we observed that formation of MG was below the detectable level (<25 nM) after incubation of ribated-BSA (Fig. 2c). Otherwise, it is possible that the generated MG from ribated-BSA could not be detected due to the rapid reaction with Arg on BSA.
Furthermore, to clarify the involvement of MG, inhibitory effect of aminoguanidine (AMG), carbonyl trapping reagent, on MG-H1 formation from ribose-gelatin was measured. As result, AMG did not inhibit the formation of MG-H1 from ribose-gelatin (Fig. 3a), whereas it significantly inhibited MG-H1 formation from the incubation of MG with gelatin (Fig. 3b), positive control for AMG study. In contrast, although TBE, which contains antioxidant such as ellagic acid, did not inhibit MG-H1 formation from the incubation of MG with gelatin (Fig. 3b), it significantly inhibited MG-H1 formation from ribose-gelatin (Fig. 3a), vice versa. In addition, MG-H1 formation was also inhibited by DTPA (Fig. 4). Therefore, we thought that MG-H1 derived from reaction of ribose and proteins was formed via the oxidative reaction. Taken together, our conclusion that MG-H1 formation from Amadori products on ribated-BSA is based on two lines of evidence; (i) formation of MG was less than detectable level (<25 nM) after incubation of ribated-BSA (Fig. 2c), and (ii) AMG did not inhibit the formation of MG-H1 from ribose-gelatin (Fig. 3a).
Since we previously demonstrated that glucosone is a major intermediate during the incubation of glucose (Fig. 6 in Nagai et al., Diabetes 51: 2833-2839, 2002), we speculated that ribosone may be generated from the incubation of ribose. In fact, a clear m/z 271, speculated as DAN-ribosone adduct, spectrum was detected after incubation of ribated-BSA, suggesting that ribosone may play a role in MG-H1 formation during incubation of ribose with proteins. However, we have not described the presence of ribosone since the structure has not yet been identified by NMR. Based on this comment, the following description was added to the revised manuscript.
Discussion section:
Our LC-QTOF system detects 25 nM MG solution (0.125 pmol /5 ml injection) as shown in Fig. 2d. Although the presence of ribose and 1,2-dicarbonyls were not measured after dialysis, we observed that formation of MG was below the detectable level (<25 nM) after incubation of ribated-BSA (Fig. 2c). Otherwise, it is possible that the generated MG from ribated-BSA could not be detected due to the rapid reaction with Arg on BSA. (Page 6, line 180-184)
Taken together, our conclusion that MG-H1 formation from Amadori products on ribated-BSA is based on two lines of evidence; (i) formation of MG was less than de-tectable level (< 25 nM) after incubation of ribated-BSA (Fig. 2c), and (ii) AMG did not inhibit the formation of MG-H1 from ribose-gelatin (Fig. 3a). (Page 6, line 168-172)
Therefore, since we previously demonstrated that glucosone is a major intermediate during the incubation of glucose under oxidative condition [27], we speculated that ri-bosone may be generated from the incubation of ribose under oxidative condition. In fact, a clear m/z 271, speculated as DAN-ribosone adduct, spectrum was detected after incubation of ribated-BSA, suggesting that ribosone may play a role in MG-H1 for-mation during incubation of ribose with proteins. (Page 6, line 190-195)
- Experiments with DTPA:
There is a study by Gensberger-Reigl et al. that reported decreased MGO formation in glucose containing peritoneal dialysis fluids when DTPA was present (doi.org/10.1007/s10719-020-09964-6). Could it be possible that the presence of DTPA leads to a reduced formation of MGO and thus, to a lower amount of MG-H1? How can you differentiate between reduced MGO formation (which might lead to lower MG-H1 levels) and direct formation of MG-H1 from the Amadori product in the ribose-BSA samples with DTPA? Why was MG analyzed in ribated-BSA, but not in ribose-BSA samples?
Response:
As Gensberger-Reigl, et al reported, MG is detected after autoclaving of glucose solution, whereas MG was below the detectable level (<25 nM) when ribated-BSA was incubated at 37°C for up to 5 days (Fig. 2c), indicating that dicarbonyls such as ribosone may be an important intermediate for MG-H1 formation from ribated-BSA. Since glucosone is generated from glucose under oxidative conditions (Fig. 6 in Nagai et al., Diabetes 51: 2833-2839, 2002), DTPA may decrease the formation of ribosone from ribose, resulting in lower MG-H1 levels.
Since MG and 3DG, precursors of MG-H1, are formed by the oxidation of glycated-proteins, we speculated that MG could also be formed by the oxidation of ribated-BSA. Therefore, we focused on ribated-BSA to confirm MG formation. Based on this comment, the following description was added in the revised manuscript.
Discussion section:
Also, since glucosone is generated from glucose under oxidative conditions [27], DTPA may decrease the formation of ribosone from ribose, resulting in lower MG-H1 levels (Page 6-7, line 195-197)
- In general, a reaction scheme that shows the chemical structures would be very helpful to follow the argumentation of the mechanistic interpretation of MG-H1 formation from ribose / the Amadori product.
Response:
As described above, we expected the formation of MG from ribated-BSA and glycated-BSA (page 5, lines 145-147 in original manuscript), but MG formation was less than detectable level (please see our response for your question #2). MG generated from ribated-BSA could not be detected due to the rapid reaction with Arg on BSA. Therefore, although we agree with this comment, formation pathway(s) leading to MG-H1 formation from ribose remains unclear. Nevertheless, we believe that this study proves for the first time that ribose generates MG-H1 by reaction with proteins.
A potential formation pathway of MG-H1 from ribose is shown in Fig. 5 and chemical structures of AGEs were shown in Fig. 1 of the revised manuscript.
- Line 34: dehydration reactions are not generally irreversible, they are rather reversible and can be irreversible in particular situations. Please express this in a more differentiated way.
Response:
We agree with the comment, the text regarding dehydration was deleted from the revised manuscript.
Introduction section:
The amino residues of proteins react non-enzymatically with reducing sugars, such as glucose, via the Maillard reaction, to form advanced glycation end-products (AGEs) through irreversible reactions such as oxidation, and condensation [3, 4]. (page 1 lines 32-34)
- Lines 47/48: You mention that ribose plays an important role in glycation in vivo based on the work from Lu et al. and Han et al. At this point, it would be helpful for the reader to get the information, which is given in lines 171/172. Alternatively, information about the actual ribose content in serum of T2DM patients would help.
Response:
Based on this comment, the ribose content in the serum of T2DM patients was added in the revised manuscript.
Introduction section:
(Control: 75.71 ± 1.83 μM, T2DM: 89.90 ± 2.40 μM) (page 2 lines 45-48)
- Figure 1:
- Why did you use different units for the y-axis of 1a and 1b? I suggest normalizing the pentosidine content as well for better comparison.
Response:
Since Pentosidine is difficult to detect by LC-QTOF, it was measured by HPLC with a fluorescence detector. Therefore, AGEs (CMA, CML, CEL and MG-H1) were normalized by protein contents to compare pentosidine as shown in Table1 (nmol/g protein). Based on this comment, the following description was added to avoid confusion.
Results section:
In addition, CMA, CML, CEL, and MG-H1 were normalized by protein contents to compare pentosidine (page 2 lines 64-65)
- The error bar is missing in 1b BSA.
Response:
The error bar in BSA control is missing because the pentosidine area level in BSA control was zero (not detected). However, its quantitative level was calculated for statistical comparison with pentosidine contents in ribose-BSA.
Based on this comment, pentosidine content in BSA control was corrected to N.D. to avoid confusion.
- Line 76/77: "MG-H1 was generated by ribated-BSA in a time-dependent manner (Figure 2a)". There is no time curve displayed in Figure 2a.
Response:
Thanks for pointing this out. "MG-H1 was generated by ribated-BSA in a time-dependent manner (Figure 2a)" has been corrected to "MG-H1 was generated by ribated-BSA in a time-dependent manner (Figure 2b)". We added the sentence regarding Figure 2a in the results section. The revised manuscript has been modified as follows.
Results section:
The generation of MG-H1 from ribated- and glycated-BSA was confirmed (Figure 2a). (page 3 lines 78-79)
- Lines 79/80: "The amount of arginine in ribated-BSA (modification ratio) did not change with incubation time (Table 2)."
In the heading to Table 2 it is written that Arg residues on ribated-BSA are reported; in the table, however, ribose-BSA is mentioned.
What do you mean by "modification rate" and how is it calculated?
I do not understand the statement in lines 79/80. How did you determine the "amount of Arg" and how can the numbers (?) of Arg residues change during incubation experiments? After incubation, BSA will still have 23 Arg residues, some of them will be modified, some of them probably not. Do the numbers of Arg residues in Table 2 represent the number of modified or unmodified Arg residues? How did you analyze the number of Arg residues? Please clarify this. It might be better to display the amount of free Arg in the samples before and after incubation analyzed by LC-QTOF (see section 4.1).
According to Uniprot entry P02769 BSA usually has 26 Arg residues. Where does this difference (26 vs. 23) come from?
Response:
Lines 79/80: As you have pointed out, this sentence has been corrected to ribated-BSA. Please see our response to question 1 from reviewer 1 for the calculation of the modification rate of Arg.
Regarding Arg contents on BSA, the sequences in the Uniprot database “P02769 BSA” are the premature form of BSA, which needs to be taken into consideration. Please see the ‘PTM / Processing section’ on P02769. We can confirm the sequences of mature BSA at chain subsections.
- Figure 2:
In chapter 4.2 only the quantification of MG-H1 is mentioned, while the amount of Amadori-compounds is displayed in Figure 2a (black bars). In addition, only MG-H1 is mentioned in the respective figure caption. Thus, I assume that the black bar represents the amount of MG-H1? Please adjust the figure legend or, if my assumption is wrong, describe the analysis of the Amadori-compounds (e.g. in the Materials and Methods section) and the results more precisely. A more detailed figure caption to Figure 2a will also help (what was quantified - MG-H1 or Amadori-compounds / sum of selected AGEs)?
How can you explain the difference (factor 10³) in concentration levels between the results that are displayed in Figure 2a and 2b (nmol/mol Arg vs. mmol/mol Arg) - if Figure 2a displays the MG-H1 content (see question above)?
Response:
We apologize for the confusion regarding chapter Chapter 4.2. Figure 2a showed MG-H1 content in ribated- or glycated- BSA, an Amadori-compounds (black bars).
The vertical axis of figure 2b (nmol/mol Arg.) has been corrected to ‘mmol/mol Arg.’. As you have pointed out, this sentence has been revised as follows.
Figure 2 section:
Formation of MG-H1 and MG from Amadori compounds. (a) Quantification of MG-H1 in ribated- or glycated-BSA for 5 days. Open bars: non-modified BSA, black bars: Amadori compounds (ribated- or glycated-BSA) (n=3); (page 3 lines 86-88)
- Lines 95ff: The inhibitory effect of TBE on the MG-H1 formation in BSA is explained and referred to the results displayed in Figure 3a. In the figure caption to Figure 3a, it is stated that the results were obtained on a ribose-gelatin system. This is inconclusive. Which system was used for the experiment displayed in Figure 3a?
Please give the concentration units for TBE and AMG in Figures 3a and 3b.
I think that it should be mentioned here, why gelatin was used instead of BSA, although the authors explain this in the discussion (lines 154/155). Otherwise, the reader cannot understand, why the protein was replaced at this point.
Response:
As you have pointed out, this sentence has been revised as follows, and we have added TBE and AMG concentration to Figure 3a and 3b.
Results section:
Since TBE, which exerts anti-oxidative activity, inhibits the formation of MG-H1 in vivo [18], the inhibitory effect of TBE on MG-H1 formation from ribose-gelatin was evaluated by monoclonal anti-MG-H1 antibody [19] as ELISA is suitable for multiple samples such as in the screening of AGE inhibitors [16]. (page 4 lines 98-101)
In addition, we incubated gelatin with ribose as the arginine content in gelatin was 3-fold higher than that of BSA, and because it could easily generate MG-H1. (page 4 lines 101-103)
Figure 3a and 3b section:
- Lines 114/115: it seems that these lines belong to the figure cation of Figure 4.
Response:
Thanks for pointing this out. DTPA has been corrected to TBE. The revised manuscript has been modified as follows.
Figure 3 legend section:
The data are presented as the mean ± S.D. (a, b) **p < 0.01, vs. control (Bonferroni test). (c) **p < 0.01, with TBE vs. without TBE control (Bonferroni test). (page 5 lines 120-121)
- Please introduce abbreviation at their first use (e.g. CMA, DAN, AMG, NaPB). Does NaPB stand for sodium phosphate buffer? If so, the term "buffer" doubled in lines 214, 215, 223, 240.
Response:
Thank you for your suggestion. We have added an abbreviations section.
Abbreviations section: (page 8 lines 300-320)
- Please comment on the stability of AGEs during acidic hydrolysis.
Response:
Since CML, CEL and pentosidine are stable, but MG-H1 and CMA are decreased by about 30 and 50%, respectively, after acid hydrolysis, addition of internal standards to analytical samples before acid hydrolysis is required. Based on your comment, the following revisions have been made in the revised manuscript.
Materials and Methods section:
To correct the variation of measured AGEs by pretreatment, standard 10 pmol [2H2] CML, [2H3] MG-H1, [2H4] CEL, (PolyPeptide Laboratories, Strasbourg, France) [13C6] CMA, 5 nmol [13C6] lysine, and [13C6] arginine (Cambridge Isotope Laboratories, Tewksbury, USA) were added to the samples, and were hydrolyzed in 6 M hydrochloric acid (HCl) at 100 °C for 18 h. (page 7 lines 230-234)
- Please review line 302ff. The consecutive numbering is interrupted.
Response:
As you have pointed out, this sentence has been revised .
- Figure 5: "Pentsidine" should be "Pentosidine".
Response:
We apologize for the spelling error; this has been corrected. The revised manuscript has been modified.
- Line 87: please give the substance concentration (amount of substance / volume) of the MG standard.
Response:
Based on this comment, ‘STD (0.005 nmol)’ has been revised as ' STD (0.005 nmol/5 mL)' in the revised manuscript.
Figure legends section:
Measurement of MG from ribated-BSA for 0, 3, 5 days (n=3) and 25 nM STD (0.125 pmol/ 5 µL injection) (page 3 lines 89-90)
